# Quantum Equilibrium Propagation: Gradient-Descent Training of Quantum Systems

**Benjamin Scellier**
Rain AI
`benjamin@rain.ai`

## Abstract

Equilibrium propagation (EP) is a training framework for physical systems that minimize an energy function. EP uses the system's intrinsic physics during both inference and training, making it a candidate for the development of energy-efficient processors for machine learning. EP has been studied in various classical physical systems, including classical Ising networks and elastic networks. We present a version of EP for quantum systems, where the energy function is the Hamiltonian's expectation value, whose minimum is reached at the ground state. As examples, we study the settings of the transverse-field Ising network and the quantum harmonic oscillator network – quantum analogues of the network models studied within classical EP.

## 1 Introduction

Machine learning (ML) is currently powered by classical digital computing. Meanwhile, fundamental research explores alternative computing paradigms to enhance ML capabilities. Quantum computing leverages the principles of quantum mechanics to encode and process information in ways that classical computers cannot, potentially handling exponentially larger amounts of information. In contrast, neuromorphic computing, taking inspiration from the brain's energy efficiency, aims to leverage analog physics and compute-in-memory platforms to significantly reduce the cost of inference and training in ML [Marković et al., 2020]. The field of 'physical learning' aims to unite these efforts by exploring the inherent physics of any physical system (whether classical or quantum) for computation, without necessarily mimicking neurons and synapses [Stern and Murugan, 2023] – see Momeni et al. [2024] for a very recent review.

Over the past decades, progress in ML research has been driven by the effectiveness of frameworks for optimizing cost functions based on the backpropagation (BP) algorithm. One challenge for the field of physical learning has been the search for frameworks that are as effective as BP, while adhering to local computation and being robust to analog noise, which are essential for efficient implementations on analog compute-in-memory platforms. In recent years, several gradient-descent training frameworks for physical systems have been proposed. For instance, Lopez-Pastor and Marquardt [2023] introduced a framework applicable to arbitrary time-reversal invariant Hamiltonian systems, and Wanjura and Marquardt [2024a] developed a method for extracting weight gradients in optical systems based on linear wave scattering. The present paper focuses on the training framework known as equilibrium propagation.

Equilibrium propagation (EP) [Scellier and Bengio, 2017] is a training framework for energy-based systems, that is systems whose physics drives their state towards the minimum of an energy function (equilibrium or steady state). EP extracts the gradients of the cost function using two equilibrium states corresponding to different boundary conditions, which are then used to locally adjust the trainable weights. EP requires only knowledge of trainable interactions, and thus is applicable in partially unknown systems too. EP has been applied to various systems, including continuous

Second Workshop on Machine Learning with New Compute Paradigms at NeurIPS 2024 (MLNCP 2024).

Hopfield networks [Scellier and Bengio, 2017], resistor networks [Kendall et al., 2020], elastic and flow networks [Stern et al., 2021], spiking networks [Martin et al., 2021], weakly connected oscillatory networks [Zoppo et al., 2022], the (classical) Ising model [Laydevant et al., 2024], and coupled phase oscillators [Wang et al., 2024]. More generally, EP applies in systems obeying variational principles [Scellier, 2021]. Recent experimental demonstrations have shown the applicability of a variant of EP called 'Coupled Learning' on hardware: Dillavou et al. [2022, 2024] built two generations of self-learning resistor networks, and Altman et al. [2024] built a self-learning elastic network. Additionally, [Yi et al., 2023] used another variant of EP in a memristor crossbar array, and Laydevant et al. [2024] used EP on D-wave to train a classical Ising machine (where they used quantum annealing to reach the ground state). Simulations have further underscored the potential of EP for ML applications: in particular, Laborieux and Zenke [2022] trained an energy-based convolutional network to classify a downsampled version of the ImageNet dataset. More broadly, [Zucchet and Sacramento, 2022] have highlighted EP's general applicability to any bilevel optimization problem (beyond physical, energy-based learning), including meta-learning [Zucchet et al., 2022].

In this work, we derive a quantum version of EP. In Quantum Equilibrium Propagation (QEP), the system is brought to the ground state of its Hamiltonian, parameterized by real-valued trainable weights, to produce a prediction. The algorithm performs gradient descent on the expectation value of an observable, which serves as the cost function to optimize. The central ingredient for translating from EP to QEP is the variational principle of quantum mechanics: the Hamiltonian's expectation value is minimized (more generally, extremized) at the Hamiltonian's ground state (more generally, eigenstates). Thus, in QEP, the Hamiltonian's expectation value represents the classical EP energy function, and eigenstates represent equilibrium states. Similar to EP, QEP only requires knowledge of trainable interactions, and has a local learning rule, which might be useful for the development of specialized quantum hardware with reduced classical overhead, where measurements of the weight gradients and adjustments of the trainable weights would be performed locally. To illustrate QEP, we study the settings of the transverse-field Ising network and the quantum harmonic oscillator network – quantum analogues of the Ising model and elastic network.

In parallel with this work, two other strongly related manuscripts exploring quantum extensions of EP have recently been published on Arxiv [Massar and Mognetti, 2024, Wanjura and Marquardt, 2024b]. Massar and Mognetti [2024] also studied EP in thermal systems, where the system settles into the minimum of the free energy functional, and demonstrated how weight gradients can be extracted solely from thermal fluctuations, while Wanjura and Marquardt [2024b] established a connection between EP and Onsager reciprocity.

## 2 Quantum Equilibrium Propagation

We present Quantum Equilibrium Propagation (QEP), an extension of EP for quantum systems. For a brief presentation of classical EP, see Appendix A. A primer on the necessary concepts of quantum mechanics is provided in Appendix B.

We consider a quantum system, serving as a 'learning machine', with a Hamiltonian $\widehat{H}(w, x)$, parameterized by trainable weights $w = (w_1, w_2, \ldots, w_M)$ and an input $x = (x_1, x_2, \ldots, x_P)$. The system's ground state, represented as $|\psi(w, x)\rangle$, satisfies:

$$\widehat{H}(w, x)|\psi(w, x)\rangle = E(w, x)|\psi(w, x)\rangle \tag{1}$$

where $E(w, x)$ denotes the ground state energy. This ground state serves to encode a prediction on a target output $y = (y_1, y_2, \ldots, y_K)$ corresponding to the supplied input $x$. We also introduce a 'cost operator' $\widehat{C}(y)$ parameterized by $y$, whose expectation value in the state $|\psi(w, x)\rangle$,

$$\langle \widehat{C}(y) \rangle_{\psi(w,x)} = \langle \psi(w, x)|\widehat{C}(y)|\psi(w, x)\rangle, \tag{2}$$

serves as the cost function. The goal is to adjust the trainable weights of the Hamiltonian to minimize this cost function. Assuming that the cost operator can be integrated in the system as an interaction Hamiltonian (interaction between the system's state $|\psi\rangle$ and target output $y$), we form the 'total Hamiltonian':

$$\widehat{H}^\beta = \widehat{H}(w, x) + \beta \widehat{C}(y), \tag{3}$$

where $\beta \in \mathbb{R}$ controls the strength of the cost interaction.

Finally, we partition the trainable weights into groups, such that the Hamiltonian derivatives $\frac{\partial \widehat{H}^\beta}{\partial w_j}$ and $\frac{\partial \widehat{H}^\beta}{\partial w_k}$ commute for any pair $w_j$ and $w_k$ within the same group. Given an input-output pair $(x, y)$ from training data, QEP proceeds as follows.

1. Set $\beta = 0$ and reach the ground state $|\psi_\star^0\rangle = |\psi(w, x)\rangle$ of $\widehat{H}^0$ (with ground state energy $E_\star^0 = E(w, x)$), characterized by

$$\widehat{H}^0|\psi_\star^0\rangle = E_\star^0|\psi_\star^0\rangle. \tag{4}$$

Measure the Hamiltonian derivatives $\frac{\partial \widehat{H}^0}{\partial w_k}$ for all trainable weights of a given group. Repeat this step for the other groups of commuting trainable weights. Repeat $T$ times and denote the measurement outcomes as $h_k^{(1)}(0), h_k^{(2)}(0), \ldots, h_k^{(T)}(0)$, for each $w_k$.

2. Set $\beta > 0$ and repeat the same step as above: reach the ground state $|\psi_\star^\beta\rangle$ of $\widehat{H}^\beta$, with ground state energy $E_\star^\beta$, characterized by

$$\widehat{H}^\beta|\psi_\star^\beta\rangle = E_\star^\beta|\psi_\star^\beta\rangle. \tag{5}$$

For each Hamiltonian derivative $\frac{\partial \widehat{H}^\beta}{\partial w_k}$, denote the $T$ measurement outcomes as $h_k^{(1)}(\beta), h_k^{(2)}(\beta), \ldots h_k^{(T)}(\beta)$.

3. Update the trainable weights $w_1, w_2, \ldots, w_M$ as

$$\Delta w_k = \frac{\eta}{\beta}\left[\frac{1}{T}\sum_{t=1}^T h_k^{(t)}(0) - \frac{1}{T}\sum_{t=1}^T h_k^{(t)}(\beta)\right], \tag{6}$$

where $\eta > 0$ is a learning rate.

The central result of QEP is that the learning rule of Eq. (6) approximates one step of gradient descent on the expectation value of the cost operator, as stated in the following result.

**Theorem 1.** *The gradient of the cost function can be approximated as*

$$\nabla_w \langle\psi(w, x)|\widehat{C}(y)|\psi(w, x)\rangle = \left.\frac{d}{d\beta}\right|_{\beta=0} \langle\psi_\star^\beta|\frac{\partial \widehat{H}^\beta}{\partial w}|\psi_\star^\beta\rangle \tag{7}$$

$$\approx \frac{1}{\beta}\left[\langle\psi_\star^\beta|\frac{\partial \widehat{H}^\beta}{\partial w}|\psi_\star^\beta\rangle - \langle\psi_\star^0|\frac{\partial \widehat{H}^0}{\partial w}|\psi_\star^0\rangle\right] \tag{8}$$

$$\approx \frac{1}{\beta}\left[\frac{1}{T}\sum_{t=1}^T h^{(t)}(\beta) - \frac{1}{T}\sum_{t=1}^T h^{(t)}(0)\right]. \tag{9}$$

Theorem 1 follows from the classical EP formula (Theorem 2 in Appendix A) and the variational principle of quantum mechanics (Lemma 3 in Appendix B). First, we comment on the differences between EP and QEP, and then we discuss the properties of EP that transfer to QEP.

Compared to the classical setting (Appendix A), the characteristics of quantum measurements affect the training procedure in several ways. First, QEP involves two levels of approximation in the estimate of the gradient of the cost function. In addition to the finite difference used to approximate the derivative $\frac{d}{d\beta}$ at $\beta = 0$, a second level of approximation is due to the probabilistic nature of quantum measurements: since a quantum measurement only gives an unbiased estimate of the expectation value, multiple measurements of the Hamiltonian derivatives $\frac{\partial \widehat{H}}{\partial w_k}$ are required to get better estimates of the weight gradients. Second, since the state of the system generally changes upon measurement of a Hamiltonian derivative, the system must be reset to its ground state after each measurement. Third, the Hamiltonian derivatives $\frac{\partial \widehat{H}}{\partial w_j}$ and $\frac{\partial \widehat{H}}{\partial w_k}$ cannot generally be measured simultaneously unless they commute – we will see, however, in Section 3 that the trainable weights can typically be partitioned into $p$ groups of commuting Hamiltonian derivatives, with small $p$ ($p = 1$ or $p = 2$ in our examples).

QEP also inherits from key features of classical EP. Suppose the total Hamiltonian of the system can be expressed as the sum of Hamiltonians corresponding to individual interactions or contributions, i.e.

$$\widehat{H} = \widehat{H}_{\text{untrainable}} + \widehat{H}_1 + \widehat{H}_2 + \cdots + \widehat{H}_M \tag{10}$$

where $\widehat{H}_k$ is the Hamiltonian of an interaction parameterized by $w_k$ (for $1 \leq k \leq M$), and $\widehat{H}_{\text{untrainable}}$ does not depend on any trainable weight. Then the Hamiltonian derivatives simplify to $\frac{\partial \widehat{H}}{\partial w_k} = \frac{\partial \widehat{H}_k}{\partial w_k}$. If the trainable weight $w_k$ is stored close to where the observable $\frac{\partial \widehat{H}_k}{\partial w_k}$ is measured, the learning rule for $w_k$ is local. Moreover, the system's Hamiltonian need not be fully known: in particular, no knowledge of $\widehat{H}_{\text{untrainable}}$ is required. Finally, similar to the classical setting where the equilibrium state need not be a minimum but only a critical point (stationary state) of the energy function, QEP only requires reaching an eigenstate of the Hamiltonian, not necessarily the ground state. One condition for Eq. (8) to hold, however, is that the nudge eigenstate $|\psi_\star^\beta\rangle$ must be obtained as a smooth deformation (adiabatic transformation) of the free eigenstate $|\psi_\star^0\rangle$ when varying the nudging parameter from 0 to $\beta \neq 0$. It remains to be seen whether this condition is necessary or can be further relaxed in practice.

## 3  Examples of Quantum Systems Compatible with QEP

Next, we present for illustration the setting of the transverse-field Ising model and quantum harmonic oscillator network.

### 3.1  Quantum Ising Model

As a first example, we consider a quantum version of the classical Ising model studied in classical EP (Appendix A.1). While a classical Ising network of $N$ classical spins can be in either of $2^N$ possible configurations, a quantum Ising network of $N$ spins exists in a superposition of these $2^N$ configurations (i.e. a linear combination with coefficients in $\mathbb{C}$). Hence the difference between the classical and the quantum settings: while the state of the classical model is described by a $N$-dimensional binary-valued vector, the quantum model's state is represented by a vector in a $2^N$-dimensional complex vector space. We denote the $d = 2^N$ basis states as $|\sigma_1\sigma_2\cdots\sigma_N\rangle$ with $\sigma_k \in \{\uparrow, \downarrow\}$ for each $k \in \{1, 2, \ldots, N\}$, e.g. $|\uparrow\uparrow \cdots \uparrow\uparrow\rangle$, $|\uparrow\uparrow \cdots \uparrow\downarrow\rangle$ and similarly for the other $2^N - 2$ basis states.

Similar to the classical Ising energy function, the Hamiltonian of a quantum Ising network has couplings between spins $J_{jk} \in \mathbb{R}$ and bias fields $h_k \in \mathbb{R}$ applied to individual spins, which we view as trainable weights. For example, the Hamiltonian of the transverse-field Ising model is given by:

$$\widehat{H}_{\text{Ising}} = -\sum_{1 \leq j < k \leq N} J_{jk} \widehat{Z}_j \widehat{Z}_k - \sum_{k=1}^{N} h_k \widehat{X}_k, \tag{11}$$

where $\widehat{Z}_k$ and $\widehat{X}_k$ are the Pauli operators, defined as follows. The Pauli $\widehat{Z}_k$ operator acts as a phase-flip operator on the $k$-th spin, according to:

$$\widehat{Z}_k |\sigma_1 \cdots \sigma_{k-1} \uparrow \sigma_{k+1} \cdots \sigma_N\rangle = +|\sigma_1 \cdots \sigma_{k-1} \uparrow \sigma_{k+1} \cdots \sigma_N\rangle, \tag{12}$$

$$\widehat{Z}_k |\sigma_1 \cdots \sigma_{k-1} \downarrow \sigma_{k+1} \cdots \sigma_N\rangle = -|\sigma_1 \cdots \sigma_{k-1} \downarrow \sigma_{k+1} \cdots \sigma_N\rangle. \tag{13}$$

The Pauli $\widehat{X}_k$ operator acts as a bit-flip operator on the $k$-th spin, according to:

$$\widehat{X}_k |\sigma_1 \cdots \sigma_{k-1} \uparrow \sigma_{k+1} \cdots \sigma_N\rangle = |\sigma_1 \cdots \sigma_{k-1} \downarrow \sigma_{k+1} \cdots \sigma_N\rangle, \tag{14}$$

$$\widehat{X}_k |\sigma_1 \cdots \sigma_{k-1} \downarrow \sigma_{k+1} \cdots \sigma_N\rangle = |\sigma_1 \cdots \sigma_{k-1} \uparrow \sigma_{k+1} \cdots \sigma_N\rangle. \tag{15}$$

In this setting, the gradients of the Ising Hamiltonian with respect to the trainable weights, required in the learning rule of Eq. (6), are given by

$$\frac{\partial \widehat{H}_{\text{Ising}}}{\partial J_{jk}} = -\widehat{Z}_j \widehat{Z}_k, \qquad \frac{\partial \widehat{H}_{\text{Ising}}}{\partial h_k} = -\widehat{X}_k. \tag{16}$$

Importantly, the Pauli $\widehat{Z}_k$ operators ($1 \leq k \leq N$) commute with one another, allowing for simultaneous measurements during training. Similarly, the Pauli $\widehat{X}_k$ operators ($1 \leq k \leq N$) commute and can be measured simultaneously. However, $\widehat{Z}_j$ and $\widehat{X}_k$ do not commute. In this example, the trainable weights can be partitioned into two groups of commuting Hamiltonian derivatives.

## 3.2 Quantum Harmonic Oscillator Network

As a second example, we consider the quantum harmonic oscillator network, a quantum analogue of the elastic network model studied in classical EP (Appendix A.2). It consists of $N$ quantum particles interacting via harmonic potentials. For clarity, we assume that the particles have one-dimensional (rather than three-dimensional) positions. Whereas the state of a classical mass-spring network is represented by the $N$-dimensional vector of positions of the particles or masses $(r_1, r_2, \cdots, r_N) \in \mathbb{R}^N$, the state of the quantum network is a superposition of all these configurations. In this setting, the state vector is a function $\psi : \mathbb{R}^N \to \mathbb{C}$ (the wave function), that assigns a complex number $\psi(r_1, r_2, \cdots, r_N)$ to each possible configuration $(r_1, r_2, \cdots, r_N)$. The corresponding Hilbert space is infinite-dimensional.

The position and momentum operators of the $i$-th particle, denoted as $\widehat{r}_i$ and $\widehat{p}_i$, are defined by their action on the wavefunction as follows:

$$(\widehat{r}_i \psi)(r_1, r_2, \cdots, r_N) = r_i \psi(r_1, r_2, \cdots, r_N), \tag{17}$$

$$(\widehat{p}_i \psi)(r_1, r_2, \cdots, r_N) = -\mathbf{i}\hbar \frac{\partial \psi}{\partial r_i}(r_1, r_2, \cdots, r_N), \tag{18}$$

where $\mathbf{i}$ is the imaginary unit ($\mathbf{i}^2 = -1$) and $\hbar$ is the reduced Planck constant. We denote as $\frac{\widehat{p}_i^2}{2m_i}$ the kinetic energy operator of the $i$-th particle, where $m_i$ is the mass, and as $V(\widehat{r}_i - \widehat{r}_j)$ the interaction potential between the $i$-th and $j$-th particles. Assuming that some of these interaction potentials are harmonic potential operator, $V(\widehat{r}_i - \widehat{r}_j) = \frac{1}{2}k_{ij}(\widehat{r}_i - \widehat{r}_j)^2$, where $k_{ij}$ is the spring constant, the Hamiltonian of the system is given by:

$$\widehat{H}_{\mathrm{QHO}} = \sum_{i=1}^{N} \frac{\widehat{p}_i^2}{2m_i} + \sum_{\text{untrainable } (i,j)} V(\widehat{r}_i - \widehat{r}_j) + \frac{1}{2} \sum_{\text{trainable } (i,j)} k_{ij}(\widehat{r}_i - \widehat{r}_j)^2, \tag{19}$$

where the $k_{ij}$'s are viewed as trainable weights.

We view this system as a 'learning machine' as follows. A subset of the particles represent 'input particles' whose positions are fixed to (classical) input values. Another subset of the particles represent 'output particles' whose position operators $\widehat{r}_i^{\mathrm{out}}$ are used as output observables, and whose measurement outcomes must match target outputs $y_i$. We use the squared error cost operator

$$\widehat{C}(y) = \sum_{i=1}^{K} \left( \widehat{r}_i^{\mathrm{out}} - y_i \widehat{I} \right)^2, \tag{20}$$

where $K$ is the number of output particles and $\widehat{I}$ is the identity operator ($\widehat{I}|\psi\rangle = |\psi\rangle$ for any $|\psi\rangle$). The expectation value of this cost operator is non-negative, and it is zero if and only if the state is an eigenstate of $\widehat{r}_i^{\mathrm{out}}$ with eigenvalue $y_i$, i.e. if and only if a measurement of $\widehat{r}_i^{\mathrm{out}}$ gives outcome $y_i$ with certainty. The term $(\widehat{r}_i^{\mathrm{out}} - y_i \widehat{I})^2$ represents a harmonic potential around $y_i$, where the $i$-th output particle experiences a restoring force that pulls it toward $y_i$. To implement the nudging Hamiltonian $\beta \widehat{C}(y)$ corresponding to this cost operator, we use $K$ springs with spring constants $k_i^{\mathrm{out}} = \beta$, connecting the $K$ output particles to $K$ 'target particles' placed at positions $y_1, \ldots, y_K$.

Finally, the QEP learning rule requires the partial derivatives of the Hamiltonian with respect to the spring constants. These are given by:

$$\frac{\partial \widehat{H}_{\mathrm{QHO}}}{\partial k_{ij}} = \frac{1}{2}(\widehat{r}_i - \widehat{r}_j)^2. \tag{21}$$

Since the $\widehat{r}_i$ operators commute with one another, all Hamiltonian derivatives can be measured simultaneously to obtain the gradients of the cost function. This example also illustrates that the

details of untrainable interactions need not be known: specifically, we do not require analytical knowledge of the Hamiltonian term

$$\widehat{H}_{\text{untrainable}} = \sum_{i=1}^{N} \frac{\widehat{p}_i^2}{2m_i} + \sum_{\text{untrainable } (i,j)} V\left(\widehat{r}_i - \widehat{r}_j\right). \tag{22}$$

## 4    Discussion

Equilibrium Propagation (EP) has been studied in various classical physical systems, including classical Ising networks and elastic networks. We have derived Quantum Equilibrium Propagation (QEP), a quantum extension of EP, and have illustrated it in quantum versions of these network models. The key conceptual bridge between EP and QEP is the variational principle of quantum mechanics, which states that the ground state of a Hamiltonian minimizes its expectation value (or more generally, its eigenstates extremize its expectation value).

QEP inherits from key features of EP. Notably, it is partially agnostic to the system's Hamiltonian, requiring only analytical knowledge of trainable interactions. Additionally, QEP employs a local learning rule. These features suggest potential benefits for developing specialized quantum computing devices that are tolerant to device variations, and where the trainable weights would be located near the location where the Hamiltonian derivatives are measured (thereby reducing the classical computational overhead). QEP thus fundamentally departs from hybrid quantum-classical approaches, where a classical computer optimizes the parameters of a parameterized quantum circuit, typically necessitating full knowledge of the circuit [Cerezo et al., 2021]. QEP is also significantly more efficient than methods that estimate partial derivatives of the loss function sequentially by perturbing each trainable weight individually [Schuld et al., 2019].

Despite these potential advantages, QEP also comes with its requirements and challenges. First, similar to classical EP, the cost operator $\widehat{C}(y)$ must be implementable as an interaction Hamiltonian, with its interaction strength controllable through the nudging parameter $\beta \in \mathbb{R}$ (see, however, Wanjura and Marquardt [2024b], where a more generic linearized nudging method is employed). Second, QEP relies on equilibration steps. While reaching the ground state of a complex Hamiltonian remains a challenging problem, QEP only necessitates reaching an *eigenstate*, which provides significantly more flexibility and may mitigate some of the practical difficulties associated with full ground-state preparation.

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

## A   Equilibrium Propagation

In this appendix, we review the Equilibrium Propagation (EP) training framework [Scellier and Bengio, 2017] and the classical Ising network and elastic network where it has been used.

EP applies in systems governed by dynamics that drive their state $s$ towards the minimum of an energy function $\mathcal{E}(s)$. These systems contain trainable weights $w = (w_1, w_2, \ldots, w_M)$ and can take an input $x = (x_1, x_2, \ldots, x_P)$ supplied as a boundary condition. We denote the corresponding energy function as $\mathcal{E}(w, x, s)$. During inference, given an input $x$, the system evolves towards its equilibrium or steady state, characterized by

$$s(w, x) = \arg\min_s \mathcal{E}(w, x, s). \tag{23}$$

The system thus implements a function $x \mapsto s(w, x)$. Training the system consists in adjusting the weights $w$ so that $s(w, \cdot)$ matches a desired input-output function. Mathematically, we use a cost function $\mathcal{C}(s(w, x), y)$ which, given an input $x$ and its associated desired output $y$, measures the accuracy of the prediction $s(w, x)$ by comparing it with $y$. Training the system can be formulated as a bilevel optimization problem [Zucchet and Sacramento, 2022]:

$$\text{minimize } \mathcal{J}(w) = \mathbb{E}_{(x,y)} \left[ \mathcal{C}(s(w, x), y) \right], \tag{24}$$

$$\text{subject to } s(w, x) = \arg\min_s \mathcal{E}(w, x, s), \tag{25}$$

where $\mathbb{E}_{(x,y)}$ represents the expectation value over input-output pairs $(x, y)$ from the training data. The conventional method to solve this problem is gradient descent on the upper-level cost function: at each step of training, an input-output pair $(x, y)$ is picked from the training data, and the trainable weights are adjusted in proportion to the gradient of the cost function: $\Delta w = -\eta \nabla_w \mathcal{C}(s(w, x), y)$, where $\eta > 0$ is a learning rate. The remaining task is to obtain or estimate the weight gradients, $\nabla_w \mathcal{C}(s(w, x), y)$, using the system's physics. This is what EP enables us to do. The central idea of EP is to view the cost function $\mathcal{C}(s, y)$ as the energy of an interaction between the state variables $(s)$ and desired outputs $(y)$, which can be incorporated into the system's energy function to form the 'total energy function',

$$\mathcal{E}^\beta(w, x, s, y) = \mathcal{E}(w, x, s) + \beta \mathcal{C}(s, y), \tag{26}$$

where $\beta \in \mathbb{R}$ is a parameter termed the 'nudging parameter' that controls the strength of this new interaction. EP proceeds in three steps:

1.  Set $\beta = 0$ and let the system settle to an equilibrium state $s_\star^0$, called the 'free state', characterized by

    $$s_\star^0 = \arg\min_s \mathcal{E}^0(w, x, s, y) = s(w, x). \tag{27}$$

    For each $k = 1, 2, \ldots, M$, measure $\frac{\partial \mathcal{E}^0}{\partial w_k}(w_1, \ldots, w_M, x, s_\star^0, y)$, i.e. the partial derivative of the energy function with respect to $w_k$.

2.  Set $\beta > 0$ and let the system reach a new equilibrium state $s_\star^\beta$, called the 'nudge state', characterized by

    $$s_\star^\beta = \arg\min_s \mathcal{E}^\beta(w, x, s, y). \tag{28}$$

    Measure again the partial derivative $\frac{\partial \mathcal{E}^\beta}{\partial w_k}(w_1, \ldots, w_M, x, s_\star^\beta, y)$ for each $k = 1, 2, \ldots, M$.

3.  Update the trainable weights $w_1, w_2, \ldots, w_M$ as

    $$\Delta w_k = \frac{\eta}{\beta} \left[ \frac{\partial \mathcal{E}^0}{\partial w_k}(w, x, s_\star^0, y) - \frac{\partial \mathcal{E}^\beta}{\partial w_k}(w, x, s_\star^\beta, y) \right], \tag{29}$$

    where $\eta > 0$ is the learning rate.

The main theoretical result of EP is that the above contrastive learning rule (29) approximates one step of gradient descent on the cost function.

**Theorem 2** (Equilibrium Propagation). *The gradient of the cost function with respect to the trainable weights can be approximated as*

$$\nabla_w \mathcal{C}(s(w,x),y) = \frac{d}{d\beta} \frac{\partial \mathcal{E}^\beta}{\partial w}(w,x,s_\star^\beta,y)\bigg|_{\beta=0} \tag{30}$$

$$\approx \frac{1}{\beta}\left[\frac{\partial \mathcal{E}^\beta}{\partial w}(w,x,s_\star^\beta,y) - \frac{\partial \mathcal{E}^0}{\partial w}(w,x,s_\star^0,y)\right]. \tag{31}$$

Theorem 2 is proved in Scellier and Bengio [2017]. One difference with traditional machine learning methods which use automatic differentiation (i.e. backpropagation) is that EP does not perform exact gradient descent on the cost function, but rather approximates it. Improved versions of EP that mitigate this problem have been proposed [Laborieux et al., 2021, Laborieux and Zenke, 2022, Scellier et al., 2024]. Next, we discuss some important features of EP.

First, in a wide range of physical systems, the contrastive learning rule of Eq. (29) is local for each trainable weight. To see this, let $w = (w_1, w_2, \ldots, w_M)$ be the set of trainable weights, and assume that the energy function is separable,

$$\mathcal{E} = \mathcal{E}_{\text{untrainable}} + \mathcal{E}_1 + \mathcal{E}_2 + \cdots + \mathcal{E}_M, \tag{32}$$

where each $\mathcal{E}_k$ (with $1 \leq k \leq M$) is the energy term of an interaction parameterized by $w_k$ (and $w_k$ only), while $\mathcal{E}_{\text{untrainable}}$ is an energy term that does not depend on any trainable weight. The energy derivatives arising in the learning rule simplify as $\frac{\partial \mathcal{E}}{\partial w_k} = \frac{\partial \mathcal{E}_k}{\partial w_k}$. If the energy term $\mathcal{E}_k$ involves only state variables spatially close to $w_k$, the learning rule for $w_k$ is local in space. Below we illustrate this property in the classical Ising model [Laydevant et al., 2024] and elastic network model [Stern et al., 2021].

Second, the system's energy function may be partially unknown. Specifically, in the above decomposition, while knowledge of the energy derivatives $\frac{\partial \mathcal{E}_k}{\partial w_k}$ is required, the untrainable energy term $\mathcal{E}_{\text{untrainable}}$ need not be analytically known. Below we illustrate this property in the elastic network model.

Last, the equilibrium states $s(w,x)$ and $s_\star^\beta$ of Eq. (23) and Eq. (28) need not be stable (i.e. minima of their repective energy functions) for Theorem 2 to hold. Theorem 2 is more generally valid when $s(w,x)$ and $s_\star^\beta$ are critical points (i.e. saddle points) of their respective energy functions, satisfying the stationary conditions

$$\frac{\partial \mathcal{E}}{\partial s}(w,x,s(w,x)) = 0, \qquad \frac{\partial \mathcal{E}^\beta}{\partial s}(w,x,s_\star^\beta,y) = 0. \tag{33}$$

where $s_\star^\beta$ is obtained as a smooth deformation of $s_\star^0$ as we gradually vary the nudging parameter from 0 to $\beta \neq 0$. See Scellier [2021, Chapter 2] for a brief discussion. So far, this feature of EP hasn't proved very useful in practice given that such equilibrium states are not stable for the system's dynamics. However, this feature may be useful in QEP (where the quantum system must reach any eigenstate of the Hamiltonian, not necessarily the ground state).

Next, we review two examples of physical systems where EP has been studied.

### A.1 Ising Network

The (classical) Ising model of coupled spins is a widely studied model that has been explored as a computing platform for machine learning, and recently explored in the context of EP [Laydevant et al., 2024]. The model consists of $N$ classical spins, characterized by their state $\sigma_k \in \{+1, -1\}$ for $1 \leq k \leq N$, representing "up" or "down" states. The state of the system is represented by the $N$-dimensional vector of spin states, $(\sigma_1, \sigma_2, \ldots, \sigma_N)$, so the state space is discrete and finite, consisting of $2^N$ possible configurations. The Ising energy function that the system seeks to minimize is defined as

$$\mathcal{E}_{\text{Ising}}(\sigma_1, \sigma_2, \ldots, \sigma_N) = -\sum_{1 \leq j < k \leq N} J_{jk}\sigma_j\sigma_k - \sum_{k=1}^N h_k\sigma_k, \tag{34}$$

where $J_{jk}$ represents the couplings between spins, and $h_k$ represents the bias fields applied to individual spins. These parameters serve as trainable weights in the model. The partial derivatives of

the energy function with respect to these trainable weights, given by

$$\frac{\partial \mathcal{E}_{\text{Ising}}}{\partial J_{jk}} = -\sigma_j \sigma_k, \qquad \frac{\partial \mathcal{E}_{\text{Ising}}}{\partial h_k} = -\sigma_k, \qquad (35)$$

involve only information that is locally available to $J_{jk}$ and $h_k$, respectively.

In terms of hardware demonstration, Laydevant et al. [2024] implemented an Ising network on the D-Wave Ising machine, employing the quantum annealing procedure of D-Wave to reach the ground state. They trained it to classify the MNIST handwritten digits using EP. They also emulated a small convolutional Ising network, using the Chimera architecture of D-Wave's chips to implement the necessary convolutional operations.

## A.2 Elastic Network

As a second example, we consider the elastic network model, studied by Stern et al. [2021] in the context of Coupled Learning (CL), a variant of EP (see Scellier et al. [2024] for a comparison of CL and EP). While Stern et al. [2021] considered networks of linear springs and used CL for training, here we also treat the case of nonlinear springs and use EP.

We consider a network of $N$ masses $m_1, m_2, \ldots, m_N$ interconnected by springs. We denote the position of mass $m_i$ as $\vec{r}_i$, and we write $\vec{r}_{ij} := \vec{r}_i - \vec{r}_j$. We denote $\mathcal{E}_{ij}(\vec{r}_{ij})$ the elastic potential energy stored in the spring between masses $m_i$ and $m_j$. We assume that some of the network springs follow Hooke's law, whose energy term is $\mathcal{E}_{ij}(\vec{r}_{ij}) = \frac{1}{2} k_{ij} (\|\vec{r}_{ij}\| - \ell_{ij})^2$, where $k_{ij}$ is the spring constant and $\ell_j$ is the spring's rest length. The state of the system is the vector of mass positions, $(\vec{r}_1, \vec{r}_2, \ldots, \vec{r}_N)$, and the total elastic energy stored in the network is given by

$$\mathcal{E}_{\text{elastic}}(\vec{r}_1, \vec{r}_2, \ldots, \vec{r}_N) = \sum_{\text{untrainable } (i,j)} \mathcal{E}_{ij}(\vec{r}_i - \vec{r}_j) + \sum_{\text{trainable } (i,j)} \frac{1}{2} k_{ij} (\|\vec{r}_i - \vec{r}_j\| - \ell_{ij})^2, \quad (36)$$

where we view the $k_{ij}$'s and $\ell_j$'s as the system's trainable weights.

This system can be used as a 'learning machine' as follows. A subset of the masses represent 'input masses' whose positions are set to given input values, and another subset of the masses represent 'output masses'. We use the squared error cost function:

$$\mathcal{C}(r^{\text{out}}, y) = \sum_{i=1}^{K} \|\vec{r}_i^{\text{out}} - \vec{y}_i\|^2, \qquad (37)$$

where $K$ is the number of output masses, $r^{\text{out}} = (\vec{r}_1^{\text{out}}, \ldots, \vec{r}_K^{\text{out}})$ is the vector of their positions, and $y = (\vec{y}_1, \ldots, \vec{y}_K)$ is the corresponding vector of desired outputs. The nudging term $\beta \mathcal{C}(r^{\text{out}}, y)$ corresponding to this cost function can be implemented using $K$ springs with spring constant $k_i^{\text{out}} = \beta$ and rest length $\ell_i = 0$, connecting the $K$ output masses to $K$ 'desired output' masses placed at positions $(\vec{y}_1, \ldots, \vec{y}_K)$.

As in the Ising model, the energy function is separable, so the learning rule is local. Specifically, the partial derivatives of the energy function with respect to these weights are given by

$$\frac{\partial \mathcal{E}_{\text{elastic}}}{\partial k_{ij}} = \frac{1}{2} (\|\vec{r}_i - \vec{r}_j\| - \ell_{ij})^2, \qquad \frac{\partial \mathcal{E}_{\text{elastic}}}{\partial \ell_{ij}} = k_{ij} (\ell_{ij} - \|\vec{r}_i - \vec{r}_j\|). \qquad (38)$$

This example also illustrates that EP is agnostic to the characteristics of untrained interactions: specifically, the energy term

$$\mathcal{E}_{\text{untrainable}} = \sum_{\text{untrainable } (i,j)} \mathcal{E}_{ij}(\vec{r}_i - \vec{r}_j), \qquad (39)$$

need not be analytically known. For such untrainable interactions, the interacting force $\vec{F}_{ij}$ between $m_i$ and $m_j$ may be any central force, of the form $\vec{F}_{ij} = -f_{ij}(\|\vec{r}_{ij}\|) \frac{\vec{r}_{ij}}{\|\vec{r}_{ij}\|}$ where $f_{ij}(\cdot)$ is an arbitrary (linear or nonlinear) characteristic. The corresponding elastic energy is $\mathcal{E}_{ij}(\vec{r}_{ij}) = \int_0^{\|\vec{r}_{ij}\|} f_{ij}(u) du$. A difference with the Ising model is that the space of possible network configurations for the elastic

network, $\mathbb{R}^{3N}$, is continuous and infinite (whereas the space of configurations of the Ising model is discrete and finite).

An experimental realization of an elastic network that learns using CL was performed by Altman et al. [2024]. In their implementation, they used the spring rest lengths $\ell_{ij}$ as trainable weights, while keeping the spring constants fixed (untrained).

# B Concepts of Quantum Mechanics

In this appendix, we present the basic concepts of quantum mechanics used in QEP. In particular, we present the variational principle of quantum mechanics, which allows us to translate from EP to QEP (Lemma 3).

The **state vector** of a quantum system, denoted $|\psi\rangle$, belongs to a complex vector space $\mathcal{H}$ equipped with an inner product $\langle \cdot | \cdot \rangle$ (specifically, a Hilbert space). For simplicity of presentation, we assume here that $\mathcal{H}$ is finite-dimensional with dimension $d$. The system's **Hamiltonian**, $\widehat{H}$, is a linear operator acting on the Hilbert space, $\widehat{H} : \mathcal{H} \to \mathcal{H}$, with the property of being self-adjoint. Due to $\widehat{H}$ being self-adjoint, its eigenvalues are real. We denote the eigenvectors of $\widehat{H}$ as $|\psi_0\rangle, |\psi_1\rangle, ..., |\psi_{d-1}\rangle$, and the associated eigenvalues as $E_0 \leq E_1 \leq \ldots \leq E_{d-1}$, such that:

$$\widehat{H}|\psi_k\rangle = E_k|\psi_k\rangle, \qquad 0 \leq k \leq d-1. \tag{40}$$

Eq. (40) is known as the time-independent Schrödinger equation. The eigenvectors $|\psi_k\rangle$ are also called the **eigenstates** of the Hamiltonian, and their associated eigenvalues $E_k$ are the energy levels. The eigenstate $|\psi_0\rangle$ with the lowest energy level is the **ground state**.

In quantum mechanics, a measurable physical quantity is represented by a self-adjoint operator, $\widehat{O} : \mathcal{H} \to \mathcal{H}$, called an **observable**. The set of possible outcomes of measuring $\widehat{O}$ is the set of eigenvalues of $\widehat{O}$, denoted $o_0, o_1, ..., o_{d-1}$, which are real due to the self-adjoint property. A peculiar aspect of quantum mechanics is that measurement outcomes are inherently probabilistic. When the system is in state $|\psi\rangle$, the probability of obtaining outcome $o_k$ upon measuring $\widehat{O}$ is given by the Born rule, $p_k = |\langle o_k|\psi\rangle|^2$, where $|o_k\rangle$ is the eigenstate associated with $o_k$, i.e. such that $\widehat{O}|o_k\rangle = o_k|o_k\rangle$. The **expectation value** of a measurement of $\widehat{O}$ when the system is in state $|\psi\rangle$ is denoted $\langle \widehat{O} \rangle_\psi$ and calculated as $\langle \widehat{O} \rangle_\psi = \sum_{k=1}^{d} p_k o_k$. Using the spectral theorem for self-adjoint operators, it can be shown that this expectation value rewrites

$$\langle \widehat{O} \rangle_\psi = \langle \psi | \widehat{O} | \psi \rangle. \tag{41}$$

In statistical terms, the expectation value represents the average result of a large number of measurements of the observable $\widehat{O}$ performed on the system in state $|\psi\rangle$.

The Hamiltonian $\widehat{H}$ is an example of an observable, with possible measurement outcomes being the energy levels $E_0, E_1, ..., E_{d-1}$. The central result that allows us to transpose EP to quantum systems is the following variational formulation of the Hamiltonian's ground state (Lemma 3). It tells us that the Hamiltonian's expectation value and the ground state can be viewed as EP's 'energy function' and 'equilibrium state', respectively.

**Lemma 3.** *The ground state $|\psi_0\rangle$ achieves the minimum of the Hamiltonian's expectation value:*

$$|\psi_0\rangle = \underset{\psi \in \mathcal{H}, \|\psi\|=1}{\arg\min} \langle \psi | \widehat{H} | \psi \rangle. \tag{42}$$

More generally, the eigenstates of the Hamiltonian $\widehat{H}$ are the critical points of the Rayleigh quotient $\psi \mapsto \frac{\langle \psi | \widehat{H} | \psi \rangle}{\langle \psi | \psi \rangle}$.

Another peculiar aspect of quantum mechanics is that the act of measuring an observable usually changes the system's state. Specifically, upon measurement of an observable $\widehat{O}$, if the outcome is $o_k$, then the system's state $|\psi\rangle$ instantaneously "collapses" to the eigenstate $|o_k\rangle$ corresponding to eigenvalue $o_k$. This principle, known as **state collapse**, implies that measuring $\widehat{O}$ a second time immediately after the first measurement will yield the same outcome $o_k$ and leave the state $|\psi\rangle = |o_k\rangle$ unchanged, in accordance with the Born rule ($p_k = \langle \psi | o_k \rangle = \langle o_k | o_k \rangle = 1$). However, state collapse has another consequence that does not exist in classical mechanics. Suppose we want to measure two observables $\widehat{O}$ and $\widehat{P}$ in state $|\psi\rangle$, aiming to obtain (unbiased) estimates of both $\langle \widehat{O} \rangle_\psi$ and $\langle \widehat{P} \rangle_\psi$. Since, in general, the system is no longer in state $|\psi\rangle$ after measuring $\widehat{O}$, we must first reset the system to state $|\psi\rangle$ before measuring $\widehat{P}$. Nonetheless, there is one notable case where it is legitimate to measure $\widehat{O}$ and $\widehat{P}$ successively without resetting the state of the system between the two measurements: this is when the two observables **commute**, i.e.

$$\widehat{O}\widehat{P} = \widehat{P}\widehat{O}. \tag{43}$$

In this case, the two operators $\widehat{P}$ and $\widehat{O}$ are simultaneous diagonalizable. Assuming for simplicity that the eigenvalues of $\widehat{O}$ are all distinct, this means that the eigenstates $|o_0\rangle$, ..., $|o_{d-1}\rangle$ of $\widehat{O}$ are also eigenstates of $\widehat{P}$. Therefore, the probability of collapsing to any eigenstate $|o_k\rangle$ is the same under each observable (given by the Born rule, $p_k = |\langle o_k|\psi\rangle|^2$), and subsequent measurements of either $\widehat{O}$ or $\widehat{P}$ will leave the state unchanged. This allows for successive measurements of $\widehat{O}$ and $\widehat{P}$ to obtain unbiased estimates of their expectation values in the initial state $|\psi\rangle$, without resetting the system between the measurements.

