# OpenReview forum: "Quantum Equilibrium Propagation: gradient-descent training of quantum systems"
_NeurIPS.cc/2024/Workshop/MLNCP — MLNCP Oral_

### Official Review · Reviewer_pHN3 · 2024-10-02
**Presents good theory but lacks numerical or experimental results**

**Rating:** 7
**Confidence:** 4

**Review:**

The authors attempt to solve quantum systems with machine learning techniques which is great field to do research. For many branches in science, it is absolutely important to understand and incorporate quantum phenomena to get meaningful results.

The paper is well-written and well-organized.

Strong points:

1.They paper develops the theory for quantum equilibrium propagation which is a quantum extension of ordinary (or classical) equilibrium propagation. The authors dis a great job explaining the details of the theory.
2. They support their theory examples where the QEO theory can be applied.

Weak points:

1. The authors do not provide any numerical results.
2. The authors use transverse field Ising model as an example, which is relatively simple quantum model. I'd like to see the proposed algorithms performance on more complicated model like anti-ferromagnetic Heisenberg model.
3. The paper could possibly benefit from a discussion on how (or if) QEP can address/circumvent the notorious 'sign-problem' that plagues the simulations of many important quantum systems.

---

### Official Review · Reviewer_gryD · 2024-10-03
**Recommendation for acceptance**

**Rating:** 8
**Confidence:** 4

**Review:**

The authors formulate a quantum version of the equilibrium propagation (EP), initially an optimization algorithm for a classical energy-based model. In addition to the algorithm formulation, the authors give examples of quantum Hamiltonian optimization setups where quantum EP (QEP) can be applied. Classical EP has been successfully applied to many classical non-von-Neumann systems in both simulation and physical experiments, and it has caught the attention of a broad research community. As such, the presented work is timely and
will interest the community. The work was also thorough and well-written. As such, I recommend acceptance of this work to MLNCP. That being said, I have a small question and comment that, if addressed, would improve the work's readability.

1) In lines 121-123, the authors clarified that the nudge eigenstate must be obtained as a smooth deformation of the free eigenstate when varying the nudging parameter. It will help readability if the authors give examples and counter-examples of the availability of smooth deformation of eigenstates.

2) In lines 198-202, the authors suggest a possible numerical simulation for QEP. Because a simulation of a 2 to 3-qubit system is fairly easy, it would help understand the performance of QEP if some simple numerical experiments could be carried out.

---

### Decision · Program_Chairs · 2024-10-10

Accept (Oral)